# International medical graduates' experiences of clinical competency assessment in postgraduate and licensing examinations: A scoping review protocol

Helen Hynes[1]*, Anél Wiese[1], Nora McCarthy[1], Catherine Sweeney[1], Tony Foley[2,3], Deirdre Bennett[1]

1 Medical Education Unit, School of Medicine, University College Cork, Cork, Ireland, 2 Department of General Practice, School of Medicine, University College Cork, Cork, Ireland, 3 Irish College of General Practitioners, Dublin, Ireland

☯ These authors contributed equally to this work.
* h.hynes@ucc.ie

**Data Availability Statement:** This is a scoping review protocol. What we are seeking to publish is

## Abstract

An international medical graduate (IMG) is a doctor who has received their basic medical qualification from a medical school located in a different country from that in which they practice or intend to practice. IMGs are known to face difficulties in their working lives, including differential attainment in assessment. The objective of this review is to map key concepts and types of evidence in academic and gray literature relating to international medical graduates' experiences of clinical competency assessment and to identify knowledge gaps on this topic by systematically searching, selecting, and synthesizing existing knowledge. All studies will relate to IMGs. The concept of interest will be IMGs' experiences of assessment. The context will be postgraduate, licensing or credentialing medical assessments of clinical competence. This review will be conducted in accordance with the Joanna Briggs Institute (JBI) methodology for scoping reviews. Seven electronic databases will be searched for literature published between 2009 and 2024: the Australian Education Index, British Education Index, ERIC, PubMed, PsycINFO, Scopus, and SocINDEX. Gray literature will be searched using Google, Google Scholar, and published reports from postgraduate training bodies and medical licensing organizations. Documents will be independently screened, selected, and extracted by two researchers using a piloted data-extraction tool. Data will be analyzed and presented in tables and in a narrative format.

**Trial resgistration: Scoping review registration:** Open Science Framework: https://osf.io/8gdm7.

## Introduction

The term 'international medical graduate' (IMG) refers to a doctor who received their basic medical qualification from a medical school located in a different country from that in which they practice or intend to practice [1, 2]. In 2021, nearly one-fifth (19%) of doctors across

the protocol for the study - the study has not yet been carried out and therefore therefore the study does not yet have any results. The only datasets thus far are the search strategy and inclusion / exclusion criteria. These are available within the manuscript and supporting files and S1 and S2.

**Funding:** The author(s) received no specific funding for this work.

**Competing interests:** The authors have declared that no competing interests exist.

OECD countries had obtained at least their basic medical qualification in another country, up from 15% a decade earlier, with the percentage of foreign-trained doctors currently exceeding 24% in Canada, 31% in the UK, 32% in Australia, 40% in Ireland, and 42% in New Zealand [3].

International medical graduates often fill in gaps in healthcare provision and are more likely to work in underserved areas or in non-training service positions [4, 5]. Many studies have shown that IMGs experience adaptation difficulties when beginning to work in a new country, including professional disorientation, integration difficulties, and barriers to training entry [1, 2, 6, 7]. A number of recent systematic and scoping reviews have examined the evidence relating to IMGs' integration difficulties and the discrimination they face in their working lives [2, 6, 7]. Common themes included inadequate professional recognition, lack of opportunities, marginalization, subtle interpersonal exclusions, stereotypes, stigma, and favoring local graduates. While many of the issues relating to unfavorable treatment may affect IMGs' performance and assessment outcomes, their experiences in relation to assessment were not explored in these reviews [2, 6, 7].

Lack of training opportunities and difficulty in passing exams are cited reasons why IMGs may be dissatisfied with work in their adopted countries [1]. There is a growing body of evidence to suggest the existence of a discrepancy in postgraduate assessment performance between IMGs and locally trained graduates [8, 9]. This gap between the attainment levels of different groups of doctors is known as differential attainment [10]. Differential attainment has been observed when comparing examination performance and career progression between IMGs, locally trained white doctors, and locally trained doctors from Black and Minority Ethnic groups [11]. It has been observed in both licensing and postgraduate examinations [12, 13]. The difference in attainment exists despite correcting for possible confounders [11].

Studies from the United Kingdom have revealed differential attainment in the assessments for membership of postgraduate training bodies when IMGs' performance is compared with UK graduates' performance [8, 12, 13]. Differential attainment has been observed in knowledge-based multiple-choice examinations [14, 15] as well as in clinical examinations [8, 12]. An independent review in 2013 found that white UK trained candidates were four times more likely to pass the Royal College of General Practitioners (RCGP) Clinical Skills Assessment than ethnic minority UK trained candidates and 14 times more likely to pass than candidates who had trained overseas [8]. This finding led the British Association of Physicians of Indian Origin to bring a judicial review against the RCGP and the General Medical Council (GMC) claiming discrimination [16]. This judicial review found that the RCGP was neither racially discriminatory nor in breach of its public sector equality duty, but it highlighted that there was a disparity in results between different groups, and that the RCGP must take action. It concluded that "*If it* [the RCGP] *does not act and its failure to act is the subject of a further challenge in the future, it may well be that it will be held to have breached its duty*" [16] [p. 9].

Similar evidence of differential attainment has been reported in the United States, Canada, Europe, and elsewhere [4, 17–19]. A study from Canada found that fewer than half of IMGs passed their certification Objective Structured Clinical Examination, while almost all (93.5%) Canadian and American graduates passed [17]. Similarly, a study from Sweden, of candidates taking an assessment at the end of internship, found that graduates from Swedish universities had a failure rate of less than 4% in contrast to graduates from other EU countries and non-EU graduates who failed at a rate of 21.2% and 41.6% respectively [18].

The reasons for differential attainment are not fully understood. Differential attainment is still evident, even when potential confounders are considered, such as pre-university attainment and socioeconomic status [20] own and parents' first language, motivation for being a doctor, study habits, living arrangements (home or away), and personality [11].

Some possible explanations for differential attainment include barriers to training that IMGs experience in their working lives, such as difficulty in accessing training positions, lack of insight into the system, lack of clarity regarding educational supervisors, discrepancies in training budgets, and access to study leave when compared with doctors in training positions [5, 21]. A GMC report from 2019 highlighted the importance of support in the working environment as a factor that promoted success in training progression. These supports included an inclusive workplace, a supportive trainer, and support to navigate the process of completing challenging professional examinations [22].

While differential attainment has been observed in knowledge-based assessment, this scoping review project will focus specifically on publications related to clinical competence assessment, including but not confined to OSCEs, workplace-based assessments, direct observation of procedures, and mini-clinical examinations. Publications related to knowledge-based assessments only will be excluded. The rationale for this is that there are many qualitative differences between the experience of undergoing clinical competency assessment and the experience of sitting knowledge-based written examinations [23], including the dynamics between candidates, examiners, real patients, and simulated patients that occur in clinical examinations but not in knowledge-based assessment. Additionally, clinical assessment mirrors the day-to-day practice of medicine more closely than knowledge-based written examinations and is therefore a very relevant measure of an IMG's adaptation and integration into a new healthcare system.

Traditionally, assessment of IMGs for licencing purposes has been based on knowledge based tests (MCQs) and tests of competence such as OSCEs [8, 10]. While traditional assessments can yield scores which are both reliable and valid [24] they offer a mere snapshot of candidate ability, and performance in real settings may vary greatly from that observed in structured simulated environments [25]. Postgraduate training assessment often makes use of workplace based assessment (WBA) methods such as mini-CEX, DOPS and evaluation of Entrustable Professional Activities (EPAs) [26, 27] which aim to evaluate the top level of Miller's pyramid (Does) [23].

In recent years there has increasing interest in programmatic assessment which involves the use of multiple different assessment components which build a picture over time to allow a more considered decision regarding the candidate passing or failing; progressing or not [25]. This could include assessment of skills such as teamwork, and professionalism which are difficult to examine using traditional means of assessment [25]. Programmatic assessment offers an authentic way in which to evaluate IMGs [27]. Thus this study will include clinical assessments in simulated and workplace based setting.

A preliminary search of PubMed, the Cochrane Database of Systematic Reviews, and *JBI Evidence Synthesis* was conducted to determine what evidence exists regarding IMGs' experiences in relation to postgraduate and medical licensing assessment. In 2015, the UK GMC commissioned a rapid review to understand differential attainment across medical training pathways [28]. This review found that most published studies focused on examination outcomes, such as pass/fail and progression/non-progression outcomes, and did not examine IMGs' experiences, opinions, or attitudes toward assessment. There have been no further reviews on this topic since 2015, and no systematic or scoping reviews on the topic of IMGs' experiences of clinical competency assessment were identified.

Therefore, the aim of this study is to document and understand what is known about IMGs' experiences of clinical competency assessment, and to identify gaps in our knowledge about this topic. The rationale for carrying out this work is to improve fairness and inclusivity for this group of doctors as they transition to working in a new country.

## Materials and methods

Scoping review methodology will be used. A scoping review is a method of knowledge synthesis that addresses an exploratory research question and aims to map key concepts, types of evidence, and gaps in research related to a specific field by systematically searching, selecting, analyzing, and synthesizing existing knowledge in both peer-reviewed and gray literature [29]. Because the aim of this review is to address a broad research question and to map the existing academic and gray evidence related to the topic, we consider a scoping review to be the most appropriate method for this study.

The proposed scoping review will follow the PRISMA Extension for Scoping Reviews (PRISMA-ScR) reporting guidelines [30]. A filled PRISMA-ScR checklist can be viewed in S1 Appendix. The six-step framework devised by Arksey and O'Malley [31] and further enhanced by Levac et al [32] will be used. These steps are: (i) identifying the research question, (ii) identifying relevant studies, (iii) study selection, (iv) charting the data, (v) collating, summarizing, and reporting the results, and (vi) consultation (optional). This review will be conducted in accordance with the JBI methodology for scoping reviews [33].

### Step 1: Identifying the research questions

The objective of this scoping review is to map key concepts and types of evidence in academic and gray literature relating to IMGs' experiences of clinical competency assessment and to identify the gaps in our knowledge on this topic by systematically searching, selecting, analyzing, and synthesizing existing knowledge.

The research questions are:

1. What literature has been published relating to the experiences of international medical graduates undertaking clinical postgraduate and licensing medical examinations?

2. What experiences do international medical graduates describe in relation to clinical postgraduate and licensing medical examinations?

3. What are the gaps in the literature relating to our knowledge and understanding of international medical graduates' experiences of clinical postgraduate and licensing medical examinations?

### Step 2: Identifying relevant studies

This scoping review will consider peer-reviewed and gray literature. Qualitative, quantitative, and mixed-methods studies will be included. Reports, reviews, theses, letters, book chapters, opinion pieces, and organizational documents will also be considered.

### Inclusion criteria

*Participants*. Studies will relate to international medical graduates. For this study, IMGs will be defined as medical doctors practicing in a country other than that in which they received their basic medical qualification. Studies relating to locally trained graduates and those that do not differentiate between IMGs and locally trained graduates will be excluded. Studies related to other healthcare professionals (not medical doctors) will also be excluded. This is intended as an international study and sources related to IMGs practicing in any country will be included.

*Concept*. The concept of interest will be IMGs' experiences of clinical competency assessment.

*Context*. The context will be postgraduate, licensing, or credentialing medical assessment that incorporates elements designed to measure clinical competence, including but not limited

to OSCEs, workplace-based assessments, direct observation of procedures, and mini-clinical examinations. Studies that relate only to knowledge-based assessments, such as multiple-choice assessments, will be excluded.

## Search strategy

The search strategy will aim to identify published and unpublished studies. An initial limited search of PubMed and Scopus was carried out to become familiar with the available evidence and to identify common keywords associated with the study population and topic. Subsequently, text words, index terms, and MESH headings from relevant articles were combined to develop a full search strategy for the British Education Index, ERIC, PubMed, Psych Info, Scopus, and Soc Index. The initial search of PubMed is provided in S2 Appendix. The search strategy, including all identified keywords and index terms, will be adapted for each database and/or information source. A forward and backward citation search will be conducted for all included sources of evidence to screen for additional studies.

Sources of unpublished studies or gray literature to be searched include Google, Google Scholar, and reports of relevant stakeholders, such as postgraduate medical training bodies and medical licensing organizations internationally.

Publications from 2009 to 2024, which are available in the full text, will be considered for inclusion. This date range was chosen to allow us to include studies regarding differential attainment, which were mainly published from 2012 onwards, while bearing in mind that postgraduate and licensing examinations are regularly reviewed and updated. Therefore, experiences from previous iterations of assessment are not relevant to the current issues faced by international medical graduates.

## Step 3: Study/source of evidence selection

Following the search of all sources, identified citations will be collated and uploaded into the Covidence systematic review software (Veritas Health Information, Melbourne, Australia). Duplicates will be removed. A pilot test will be conducted to trial the inclusion criteria. To minimize information bias, no language restrictions will be applied. From the initial searches, we believe that the number of non-English language papers will be low. If the need arises, the authors will draw on the available resources from their university and or use Google Translate [34] and DeepL [35] software to translate relevant papers from other languages.

After the pilot, three reviewers will use the inclusion and exclusion criteria (Table 1) to screen the titles and abstracts of all citations, with two researchers independently screening each item. Thereafter, full texts will be retrieved for sources that were identified as potentially relevant, and these will be assessed in detail against the inclusion criteria by two of the three independent reviewers. Sources that do not meet the inclusion criteria for full-text review will be removed. The reasons for exclusion will be recorded and reported in the scoping review. If any disagreements arise between the reviewers at any stage of the selection process, these sources will be discussed, and if necessary, an additional reviewer or reviewers will be involved in resolving the issue. The results of the search and the study inclusion process will be documented in the final scoping review using the Preferred Reporting Items for Systematic Reviews and Meta-Analyses extension for Scoping Review (PRISMA-ScR) flow diagram [30].

## Step (iv): Charting the data

A data extraction tool specific to the research questions will be developed and piloted by the reviewers. Three independent reviewers will use the data extraction tool to extract and record the data from the papers. The extracted data will include specific details about the study

**Table 1. Inclusion and exclusion criteria.**

|  | Inclusion | Exclusion |
|---|---|---|
| Population | Data source relating to International Medical Graduates (medical doctors) | Sources relating to locally trained medical graduates. |
|  |  | Sources which do not differentiate where the graduates trained. |
|  |  | Sources relating to other health care professionals (not doctors). |
| Concept | Sources relating to the experiences of the participants | Sources which do not refer to the experiences of participants |
| Context | Sources relating to postgraduate medical examinations. | Sources relating to undergraduate medical examinations. |
|  | Sources relating to licencing or credentialing medical examinations. |  |
|  | Sources relating to clinical competence assessment (including but not confined to OSCEs, Workplace based assessments, Direct Observation of Procedures, Mini-Clinical Examinations) | Sources relating only to knowledge-based assessment (e.g., MCQs) |
| Types of sources | Qualitative, quantitative, or mixed methods studies; 'grey' literature such as reports, reviews, theses, letters, book chapters, opinion pieces, and organisational documents. |  |
|  | Published between 2009–2024. | Published prior to 2009 |

participants, concepts, contexts, study methodologies, and key findings relevant to the review questions.

The draft extraction tool can be viewed in S3 Appendix. This will be modified and revised as necessary during the process of extracting data from each of the included sources of evidence. All modifications will be described in the scoping review. To reduce the possibility of error or bias, two of the three reviewers will independently extract the data from each source. Disagreements between the reviewers will be resolved through discussion or with the involvement of an additional reviewer. In cases where essential data are lacking, the authors of the respective papers will be contacted to request the required information.

## Step (v): Collating, summarizing, and reporting the results

As per the JBI Manual for Evidence Synthesis 2020 Guideline [33], the results will be presented as both diagrams and tables, with a descriptive summary discussing how the results relate to the review objectives and questions. The summary will also identify possible areas for further research on this topic. The results will be categorized according to the key themes identified and will relate back to the specific review objectives on the experiences of international medical graduates in relation to postgraduate and licensing clinical competency assessments.

## Step (vi): Consultation

Experienced researchers will advise throughout the process including an expert in assessment who is also an international medical graduate.

This protocol has been registered and published on the Open Science Framework (available at: https://osf.io/8gdm7).

## Discussion

The strengths of this study lie in the chosen methodology. The search strategy has been designed to be inclusive, with searches of multiple databases, gray literature and reports from

the relevant medical licensing and assessment bodies. Adhering to the JBI methodological framework for scoping reviews [33] will allow for a broad exploration of the research landscape. The use of the PRISMA ScR reporting guidelines [30] will ensure transparency at all stages in the reporting process.

A potential limitation of the study may be the difficulty in obtaining translations of sources that are not published in the English language. However, the researchers are based in a University with a large multinational faculty and global research links and every effort will be made to obtain translations of all relevant sources. As with all reviews, there is the potential for publication bias in the studies included. This will be considered and discussed in the final review.

The findings of this scoping review will be disseminated via peer reviewed journal publication and presentation at medical conferences.

This scoping review will be part of a larger piece of work which aims to explore the issues and difficulties faced by international medical graduates in relation to clinical competency assessment. While there are published reviews of the issues faced by IMGs in relation to working and acclimatization in a new country, these do not deal specifically with issues related to assessment. This work will add to the body of existing knowledge in this field. Due to the large number of IMGs now staffing health services across the globe, we believe that this review is timely and that it has the potential to suggest ways to improve assessment for international and local medical graduates.

## Supporting information

**S1 Appendix. Preferred Reporting Items for Systematic reviews and Meta-Analyses extension for Scoping Reviews (PRISMA-ScR) checklist.**
(DOCX)

**S2 Appendix. Search strategy.** Search Strategy for PubMed (National Library of Medicine)—Search conducted on Jan 20th, 2024.
(DOCX)

**S3 Appendix. Data extraction tool.**
(DOCX)

## Acknowledgments

The authors acknowledge the contribution of Virginia Conrick, a librarian at University College Cork, for her assistance in devising the search strategy.

## Author Contributions

**Conceptualization:** Helen Hynes, Anél Wiese, Nora McCarthy, Tony Foley, Deirdre Bennett.

**Methodology:** Helen Hynes, Anél Wiese, Tony Foley, Deirdre Bennett.

**Project administration:** Helen Hynes.

**Supervision:** Tony Foley, Deirdre Bennett.

**Writing – original draft:** Helen Hynes.

**Writing – review & editing:** Helen Hynes, Anél Wiese, Nora McCarthy, Catherine Sweeney, Tony Foley, Deirdre Bennett.

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
