## [Decision Letter · Decision Letter 0]

25 Jul 2024

PONE-D-24-19584International medical graduates’ experiences of clinical competency assessment in postgraduate and licensing examinations: a scoping review protocol.PLOS ONE

Dear Dr. Hynes,

Thank you for submitting your manuscript to PLOS ONE. After careful consideration, we feel that it has merit but does not fully meet PLOS ONE’s publication criteria as it currently stands. Therefore, we invite you to submit a revised version of the manuscript that addresses the points raised during the review process.

 International medical graduates are essential to developed countries and their assessment for practicing in the host country is often under debate. Most assessments focus on competency rather than performance or Entrustable Professional activities, and often do not assess communication and teamwork skills. There is increasing interest in performance-based and programmatic assessments, as well as the publication of composite reliability of workplace-based assessments. This signals a potential shift towards a performance-based approach in the assessment of international medical graduates. Please explain problem statement in couple of paragraph and signify the rational behind the study.

We look forward to receiving your revised manuscript.

Kind regards,

Muhammad Shahzad Aslam, Ph.D.,M.Phil., Pharm-D

Academic Editor

PLOS ONE

Journal Requirements:

Additional Editor Comments:

International medical graduates are essential to developed countries and their assessment for practicing in the host country is often under debate. Most assessments focus on competency rather than performance or Entrustable Professional activities, and often do not assess communication and teamwork skills. There is increasing interest in performance-based and programmatic assessments, as well as the publication of composite reliability of workplace-based assessments. This signals a potential shift towards a performance-based approach in the assessment of international medical graduates. Please explain problem statement in couple of paragraph and signify the rational behind the study.

Reviewers' comments:

Reviewer's Responses to Questions

**Comments to the Author**

1. Does the manuscript provide a valid rationale for the proposed study, with clearly identified and justified research questions?

Reviewer #1: Yes

Reviewer #2: Yes

2. Is the protocol technically sound and planned in a manner that will lead to a meaningful outcome and allow testing the stated hypotheses?

Reviewer #1: Yes

Reviewer #2: Yes

3. Is the methodology feasible and described in sufficient detail to allow the work to be replicable?

Reviewer #1: Yes

Reviewer #2: Yes

4. Have the authors described where all data underlying the findings will be made available when the study is complete?

Reviewer #1: Yes

Reviewer #2: Yes

5. Is the manuscript presented in an intelligible fashion and written in standard English?

Reviewer #1: Yes

Reviewer #2: Yes

6. Review Comments to the Author

You may also provide optional suggestions and comments to authors that they might find helpful in planning their study.

Reviewer #1: The manuscript is well-presented, and its objectives are clearly outlined. However, it is not clear whether the authors intend to conduct a global study or if the scope is limited to a specific region. This clarification is particularly pertinent to the scoping of the grey literature search. The authors are to consider specifying whether the grey literature search will cover global "postgraduate training bodies and medical licensing organizations" or be confined to specific regions. Additionally, the authors to consider outlining their approach to mitigating information bias, particularly regarding the translation of non-English papers.

Reviewer #2: International medical graduates are vital to all developed countries and are growing in numbers. Their assessment to get a licence to practise in the host country is often subject to much debate. Most assessments are based on competency and not performance or Entrustable Professional activities. Communication and teamwork skills are not assessed. There are models of performance-based assessment and there is a growing interest in programmatic assessment (Nair B , Moonen van Loon J, et al MJA, 2017). Composite reliability of workplace based assessments have been published too . This may be the future of IMG assesmsent

7. PLOS authors have the option to publish the peer review history of their article (what does this mean?). If published, this will include your full peer review and any attached files.

Reviewer #1: No

Reviewer #2: **Yes: **Balakrishnan R Nair

---

## [Author Response · Author response to Decision Letter 0]

29 Aug 2024

Dear Dr Aslam,

We would like to thank you and the PLOS ONE reviewers for your useful comments. We are very pleased that the reviewers consider the paper to be well presented with clearly outlined objectives. As regards the clarifications and suggestions, we will outline below each of the points made and our responses to same.

Reviewer #1: 

1. “It is not clear whether the authors intend to conduct a global study or if the scope is limited to a specific region. This clarification is particularly pertinent to the scoping of the grey literature search.” 

• This is intended as an international study and sources relating to International Medical Graduates (IMGs) practicing in any country will be included. Please see the clarification of this point on lines 206 and 207 of the revised paper (Manuscript).

o “This is intended as an international study and sources related to IMGs practicing in any country will be included.”

2. “The authors are to consider specifying whether the grey literature search will cover global "postgraduate training bodies and medical licensing organizations" or be confined to specific regions.”

• The grey literature search will cover global stakeholders. We feel this is an important aspect of the study because the phenomenon of doctors migrating is a truly global issue. Please see edit on line 230.

o “Sources of unpublished studies or gray literature to be searched include Google, Google Scholar, and reports of relevant stakeholders, such as postgraduate medical training bodies and medical licensing organizations internationally.” 

3. “Additionally, the authors to consider outlining their approach to mitigating information bias, particularly regarding the translation of non-English papers.”

• We believe that given the international scope of this study, and the fact that international medical graduates originate from and work in a multitude of countries around the world, that to limit the search to papers in the English language would risk information bias. We have therefore decided not to place any language limitations on the sources included in the review. We will make every effort to translate sources which are not in the English language using a combination of online software (Google Translate and DeepL) and the many language resources which are available to us in our university. Please see lines 242-246.

o “To minimize information bias, no language restrictions will be applied. From the initial searches, we believe that the number of non-English language papers will be low. If the need arises, the authors will draw on the available resources from their university and or use Google Translate35 and DeepL36 software to translate relevant papers from other languages.”

Reviewer #2: 

1. “International medical graduates are vital to all developed countries and are growing in numbers. Their assessment to get a licence to practise in the host country is often subject to much debate. Most assessments are based on competency and not performance or Entrustable Professional activities. Communication and teamwork skills are not assessed. There are models of performance-based assessment and there is a growing interest in programmatic assessment (Nair B , Moonen van Loon J, et al MJA, 2017). Composite reliability of workplace based assessments have been published too . This may be the future of IMG assessment.”

• We acknowledge the importance of programmatic assessment in postgraduate medical assessment and its potential in the authentic assessment of performance of IMGs. We have added a discussion of programmatic assessment in the introduction section. Please see lines 139-145.

o “In recent years there has increasing interest in programmatic assessment which involves the use of multiple different assessment components which build a picture over time to allow a more considered decision regarding the candidate passing or failing; progressing or not. 26 This could include assessment of skills such as teamwork, and professionalism which are difficult to examine using traditional means of assessment. 26 Programmatic assessment offers an authentic way in which to evaluate IMGs. 28 Thus this study will include clinical assessments in simulated and workplace based setting.”

Editor:

1. “International medical graduates are essential to developed countries and their assessment for practicing in the host country is often under debate. Most assessments focus on competency rather than performance or Entrustable Professional activities, and often do not assess communication and teamwork skills. There is increasing interest in performance-based and programmatic assessments, as well as the publication of composite reliability of workplace-based assessments. This signals a potential shift towards a performance-based approach in the assessment of international medical graduates. Please explain problem statement in couple of paragraph and signify the rationale behind the study.”

• We would like to thank the editor for raising this issue. We believe that programmatic assessment has great potential both in the assessment of postgraduate trainees and also for assessment of IMGs who are already working in a supervised capacity. We have added a brief description of the nature and benefits of workplace based assessment and programmatic assessment of medical practitioners including IMGs.

Please see lines 130-145.

o “Traditionally, assessment of IMGs for licencing purposes has been based on knowledge based tests (MCQs) and tests of competence such as OSCEs.8,10 While traditional assessments can yield scores which are both reliable and valid 25 they offer a mere snapshot of candidate ability, and performance in real settlings may vary greatly from that observed in structured simulated environments. 26 Postgraduate training assessment often makes use of workplace based assessment (WBA) methods such as mini-CEX, DOPS and evaluation of Entrustable Professional Activities (EPAs)27,28 which aim to evaluate the top level of Miller’s pyramid (Does). 24 

In recent years there has increasing interest in programmatic assessment which involves the use of multiple different assessment components which build a picture over time to allow a more considered decision regarding the candidate passing or failing; progressing or not. 26 This could include assessment of skills such as teamwork, and professionalism which are difficult to examine using traditional means of assessment. 26 Programmatic assessment offers an authentic way in which to evaluate IMGs. 28 Thus this study will include clinical assessments in simulated and workplace based setting.”

• We have added in the rationale for the study in lines 157-160.

o “Therefore, the aim of this study is to document and understand what is known about IMGs’ experiences of clinical competency assessment, and to identify gaps in our knowledge about this topic. The rationale for carrying out this work is to improve fairness and inclusivity for this group of doctors as they transition to working in a new country.”

We would like to thank the editors for their thorough review of our manuscript and for the opportunity to publish in PLOS ONE. Your insightful feedback has been very helpful in enhancing the clarity of the protocol. 

Yours sincerely,

Dr Helen Hynes,

Senior Lecturer in Medical Education,

Director, 5 Year Medical Programme,

School of Medicine, University College Cork

---

## [Decision Letter · Decision Letter 1]

9 Sep 2024

International medical graduates’ experiences of clinical competency assessment in postgraduate and licensing examinations: a scoping review protocol.

PONE-D-24-19584R1

Dear Dr. Hynes,

We’re pleased to inform you that your manuscript has been judged scientifically suitable for publication and will be formally accepted for publication once it meets all outstanding technical requirements.

Kind regards,

Muhammad Shahzad Aslam, Ph.D.,M.Phil., Pharm-D

Academic Editor

PLOS ONE

Additional Editor Comments (optional):

Reviewers' comments:

Reviewer's Responses to Questions

**Comments to the Author**

1. Does the manuscript provide a valid rationale for the proposed study, with clearly identified and justified research questions?

Reviewer #2: Yes

2. Is the protocol technically sound and planned in a manner that will lead to a meaningful outcome and allow testing the stated hypotheses?

Reviewer #2: Yes

3. Is the methodology feasible and described in sufficient detail to allow the work to be replicable?

Reviewer #2: Yes

4. Have the authors described where all data underlying the findings will be made available when the study is complete?

Reviewer #2: Yes

5. Is the manuscript presented in an intelligible fashion and written in standard English?

Reviewer #2: Yes

6. Review Comments to the Author

You may also provide optional suggestions and comments to authors that they might find helpful in planning their study.

Reviewer #2: The authors have revised the paper based on the comments from the reviewers . I am satisfied by the responses . The article reads well now

7. PLOS authors have the option to publish the peer review history of their article (what does this mean?). If published, this will include your full peer review and any attached files.

Reviewer #2: **Yes: **Balakrishnan R Nair

---

## [Editor Report · Acceptance letter]

11 Oct 2024

PONE-D-24-19584R1 

PLOS ONE

Dear Dr. Hynes, 

I'm pleased to inform you that your manuscript has been deemed suitable for publication in PLOS ONE. Congratulations! Your manuscript is now being handed over to our production team.

Kind regards, 

on behalf of

Dr. Muhammad Shahzad Aslam 

Academic Editor

PLOS ONE